# Evaluation of the performance of large language models in responding to medical questions related to multiple sclerosis: A case study of large language models including ChatGPT, Gemini, Grok and Copilot

Meisam Dastani[1], Mohammad Shayan Sajjadi[2], Bassem Yamout[3], Melika Arab Bafrani[4], Amirreza Nasirzadeh[5]*

1 Infectious Diseases Research Center, Gonabad University of Medical Sciences, Gonabad, Iran,
2 Student Research Committee, Gonabad University of Medical Sciences, Gonabad, Iran, 3 Neurology Institute, Harley Street Medical Center, Abu Dhabi, United Arabia Emirates, 4 Faculty of Medicine, Tehran University of Medical Sciences, Tehran, Iran, 5 Department of Medical-Surgical Nursing, Faculty of Nursing, Nursing Research Center, Gonabad University of Medical Sciences, Gonabad, Iran

* nasirzadeharnz@gmail.com

## Abstract

### Objective

To evaluate and compare the performance of four publicly available large language models—ChatGPT, Gemini, Copilot, and Grok—in answering medical questions related to Multiple Sclerosis, focusing on accuracy, transparency, and clinical actionability.

### Methods

Four publicly available large language models (ChatGPT, Gemini, Grok, and Copilot) were selected based on accessibility and their ability to respond to medical questions. A total of 25 questions—five for each of the five key domains (diagnosis, treatment, prevention, disease control, and disease management)—were developed. The responses generated by the models were evaluated using the DISCERN-AI and NLAT-AI assessment tools.

### Results

The evaluation of four AI chatbots—ChatGPT, Gemini, Copilot, and Grok—on multiple sclerosis (MS) content revealed clear differences in quality and consistency. According to DISCERN-AI criteria, Gemini achieved the highest overall quality, excelling in relevance, transparency, balance, and acknowledgment of uncertainty. Grok ranked second, showing generally balanced results with slightly lower scores than Gemini. ChatGPT exhibited strong yet uneven performance, with particular

**Data availability statement:** All relevant data are within the manuscript and its Supporting Information files.

**Funding:** The author(s) received no specific funding for this work.

**Competing interests:** The authors have declared that no competing interests exist.

weaknesses in content addressing vulnerable populations. Copilot demonstrated the weakest overall performance, with consistently lower scores across nearly all criteria.

## Conclusions

Gemini demonstrated the strongest and most consistent performance across all domains, followed by Grok with slightly lower but balanced results. ChatGPT showed strong yet uneven outcomes, with weaknesses in addressing vulnerable populations. Copilot ranked lowest, consistently underperforming across metrics. These findings highlight significant differences among large language models in generating accurate and clinically relevant responses for multiple sclerosis, underscoring the importance of considering each model's strengths and limitations in healthcare applications.

## 1. Introduction

In recent years Large Language Models (LLMs) represent a major progress on artificial intelligence (AI) and natural language processing (NLP). By training large deep neural networks on huge textual corpora, these models have developed the ability to understand and generate human-like text. Their primary capabilities are answering questions, translation, text summary, and text generation [1]. Thus, the use of LLMs is becoming widely accepted, and tools like Bard, Bing, and ChatGPT OpenAI recently built, allow users to use these models to access a variety of services [2]. According to some researchers, these models may well be the next generation of search engines and find wide use across software landscapes [3]. Initial inspection suggested these models are good at semantic and syntactic comprehension of natural language [4], and can achieve high efficiency over various natural language processing (NLP) tasks. Not only that, LLMs have demonstrated strong results on answering questions about mathematics, science, programming, logical reasoning, and even the humanities [5,6]. Due to their enormous potential to augment and revolutionize traditional approaches in many application areas, LLMs have received great attention recently [7].

In the domain of healthcare, ChatGPT has been presented as an exemplar, exhibiting promising skills in text-based human-like conversation [8]. In addition to ChatGPT, other publicly available large language models have been developed, including Gemini, Copilot, and Grok. Gemini is a multimodal LLM capable of processing text, images, and code, providing broad reasoning capabilities across multiple domains [9]. Copilot, originally designed to assist programming tasks, also offers general text-generation capabilities that can be applied in medical question answering [9]. Grok is designed for real-time conversational AI and leverages retrieval-augmented generation (RAG) to provide up-to-date and contextually relevant responses [10]. These properties have enabled experimental use cases where the system is utilized for responding to medical questions and generating accurate medical content. Moreover, the model has demonstrated successful achievements in disease diagnosis [11], treatment recommendation [12], patient education [13] and

medical image analysis [1]. Large language models such as GPT-3.5 and GPT-4 are capable of analyzing and integrating vast volumes of medical literature and patient data, which may help alleviate the burden of information overload faced by healthcare professionals [14]. Such a feature aligns nicely with well-known, low-resource healthcare systems and long patient wait times, possibly fostering the use of LLM-based chatbots like ChatGPT as an alternative consultation to highly-skilled experts [15,16]. However, there are still serious obstacles to the application of LLMs in some sensitive domains like healthcare. The challenges of ethics, patient privacy and security, and the potential ability of models to perpetuate or outright cause harm through their outputs are among these concerns [17,18].

Despite the recent breakthroughs of large language models, it is still challenging to convincingly interpret a model's performance in specific domains such as medicine. In particular, their ability to answer medical questions around specific conditions as Multiple Sclerosis (MS) merits in-depth evaluation. MS is a common, worldwide, debilitating neurological condition that affects millions of people annually. Timely and correct diagnosis and treatment of it is very important. MS is an important public health issue, which warrants special attention, and early detection and effective treatment is instrumental in the management of the condition [19].

Thus, our purpose in this paper is to assess the ChatGPT and other large language models to answer medical questions associated with MS. Due to the complexity of the diagnosis, treatment of this disease, and, more importantly, how essential it is for the patients to have access to accurate and complete information, in this research the accuracy, reliability, and completeness of generated responses for subjects related to MS are evaluated of these models. It also aims to explore the pros and cons of these models in providing medical information, and whether there is any potential hazard brought by the incomplete or inaccurate outputs. The results of this study will inform techniques to improve the use of large language models for health care and other potential domains, contributing strategies to improve their efficacy for tackling complex diseases, including MS.

## 2. Methods

In this study, the population that was investigated included public LLMs. Criteria for inclusion were public availability, and ability to answer research questions, of the models. Using these, four large language models—ChatGPT, Gemini, Grok and Copilot—were chosen as exemplars. The accessibility and ability to output textual response of these models were the reason they were selected.

### 2.1. Questions design

Medical questions were formulated based on patients' informational needs and findings from previous research in the field [19–21]. These questions were reviewed by the research team in collaboration with two certified and experienced neurologists and one MS nurse specialist involved in the direct care of MS patients. They were categorized into five key domains related to multiple sclerosis (MS): (1) diagnosis, (2) treatment, (3) prevention, (4) disease control, and (5) disease management. A total of 25 questions were designed, with five questions assigned to each domain.

### 2.2. Data collection

The questions were translated into English prompts based on the Meskó 2023 Prompt Engineering Guidelines [22]. All prompts were written in clear, formal English to ensure consistency across models and minimize variability caused by differences in phrasing. Additionally, prompts were adapted to match the general public's health literacy level, ensuring accessibility while maintaining comparability of responses between models. Each prompt was submitted separately to four AI models—ChatGPT, Gemini, Grok, and Copilot—on July 30, 2025. All responses were collected and saved systematically. For unbiased analysis, the responses were anonymized and labeled with numerical codes before qualitative and comparative evaluation.

## 2.3. Evaluation

To assess the quality of the AI-generated responses, two validated evaluation tools were employed: the DISCERN-AI and the Natural Language Assessment Tool for AI (NLAT-AI). Both instruments have been previously applied in the evaluation of language model outputs in health-related contexts and have demonstrated acceptable validity and reliability [18,23]. The DISCERN-AI tool is a modified version of the original DISCERN instrument, which is widely used to evaluate the quality of treatment information in healthcare. Individual DISCERN-AI items (items 1–6) were rated using categorical responses ("Yes", "Partially", "No"), reflecting full, partial, or absent fulfillment of each criterion. The overall quality of each AI-generated response was subsequently determined using DISCERN question 7, which classifies outputs as low, moderate, or high quality based on the distribution of item-level scores, in accordance with the original DISCERN scoring guidelines. [24].

The NLAT-AI tool was used to complement this assessment, focusing on five key criteria: accuracy, safety, appropriateness, actionability, and effectiveness. Each criterion was scored on a 5-point Likert scale, providing a multidimensional evaluation of the clinical utility and communicative quality of each AI response [17]. In this study, DISCERN-AI was primarily used to evaluate the overall quality and general reliability of the AI-generated responses, whereas NLAT-AI was employed to provide a domain-specific assessment, measuring the performance of each model across five key areas: diagnosis, treatment, prevention and control, disease management, and responses to children and vulnerable populations. By combining these two instruments, we ensured both a comprehensive overall evaluation and a detailed analysis of performance within specific medical domains.

All prompts and responses generated by the selected language models (ChatGPT, Gemini, Grok, and Copilot) were anonymized and coded to remove identifying features. These materials, along with detailed descriptions of the tools and models, were provided to an independent evaluator. The evaluator was a board-certified neurologist with expertise in clinical neurology and academic research and was not involved in the study design or data collection. The evaluated questions were general and guideline-based, and assessments were grounded in established clinical knowledge, minimizing subjective judgment. The decision to use a single evaluator was based on the exploratory and case-study nature of this investigation and is consistent with prior study evaluating large language models in medical and clinical contexts [18]. After completing a structured review of all responses, the evaluator's assessments were compiled and analyzed using descriptive statistical methods. In this study, DISCERN-AI and NLAT-AI were used as complementary evaluation frameworks addressing distinct but related aspects of LLM performance. DISCERN-AI was primarily applied to assess the overall quality, transparency, and reliability of the generated health information from a patient education perspective. In contrast, NLAT-AI was used to capture domain-specific and clinically relevant performance, focusing on accuracy, safety, appropriateness, actionability, and effectiveness across multiple MS-related clinical domains. This dual-metric approach enabled a more comprehensive and multidimensional evaluation of LLM outputs.

## 2.4. Ethical statement

The study protocol was reviewed and approved by the Research Ethics Committees of Gonabad University of Medical Sciences (Approval code: IR.GMU.REC.1404.041; Approval date: July 29, 2025). This study did not involve human participants or use any personal or patient data. All procedures were conducted in accordance with relevant guidelines and regulations, and the Ethics Committee confirmed that no informed consent was required. The authors would like to thank the Research Ethics Committees of Gonabad University of Medical Sciences for their guidance and support during the approval process of this study.

## 3. Result

Table 1 shows the results of the evaluation of responses from four AI chatbots—ChatGPT, Gemini, Copilot, and Grok—on the topic of multiple sclerosis (MS) based on the DISCERN-AI criteria.

**Table 1. Assessment of chatbot performance in answering MS-related medical questions using the DISCERN-AI criteria.**

| Criteria | ChatGPT | Copilot | Gemini | Grok |
|---|---|---|---|---|
| Relevance of information | Partially | Partially | Yes | Partially |
| Information sources | No | Yes | Yes | No |
| Date of information production | No | Yes | Yes | No |
| Balance and impartiality | Partially | Yes | Yes | Partially |
| Additional sources | No | Partially | Yes | No |
| Indication of uncertainty | Yes | Partially | Yes | Yes |
| Overall quality | Medium | Medium | High | Medium |

The data in Table 1 reveals that on the "Relevance of information" metric, AI chatbots ChatGPT, Copilot, and Grok performed similarly at a moderate level. Only Gemini demonstrated higher accuracy in connecting content to the topic.

Significant differences were observed on the "Information sources" and "Date of information production" metrics. Both Gemini and Copilot provided the highest level of transparency and up-to-datedness, whereas ChatGPT and Grok showed a noticeable weakness in this area.

Regarding "Balance and impartiality," Gemini and Copilot again outperformed the other two models with high performance. However, on the "Additional sources" metric, Gemini achieved the highest performance, and Copilot managed to provide sources at a moderate level, while ChatGPT and Grok performed poorly on this indicator.

On the "Indication of uncertainty" metric, ChatGPT, Gemini, and Grok had the highest performance, while Copilot's performance was moderate. Furthermore, on the "Overall quality" index, Gemini's performance was rated "high," and the other three models (ChatGPT, Copilot, and Grok) were all evaluated as "moderate". Table 2 summarizes the descriptive statistics for both scales across 4 LLMs.

These radar charts provide a visual representation of the performance of each chatbot model across different domains and based on the aforementioned indicators. For example, in each chart, each colored line represents a domain (such as Diagnostic or Treatment), and the points on these lines show the score obtained by the chatbot on that specific indicator (such as Accuracy or Safety) within that domain. The larger the area covered by a chatbot in these charts, the better its overall performance is evaluated.

Based on the radar charts in Fig 1, the data indicates that the Gemini model has a uniform and strong performance across all evaluated domains and indicators. Its chart is a large, symmetrical polygon, which shows that its scores on all metrics—Accuracy, Safety, Appropriateness, Actionability, and Effectiveness—are close to the maximum across the different domains (Diagnostic, Treatment, Prevention and Control, Disease Management, and Children and Vulnerable Populations). This demonstrates its high stability and quality.

**Table 2. Descriptive statistics for NLAT and DISCERN by LLM.**

| Model | NLAT Mean | NLAT SD | NLAT Median | DISCERN Mean | DISCERN SD | DISCERN Median |
|---|---|---|---|---|---|---|
| ChatGPT | 3.20 | 1.04 | 3.0 | 1.71 | 0.76 | 2.0 |
| Gemini | 4.40 | 0.50 | 4.0 | 3.00 | 0.00 | 3.0 |
| Copilot | 3.44 | 0.71 | 4.0 | 2.43 | 0.53 | 2.0 |
| Grok | 4.20 | 0.71 | 4.0 | 1.57 | 0.53 | 2.0 |

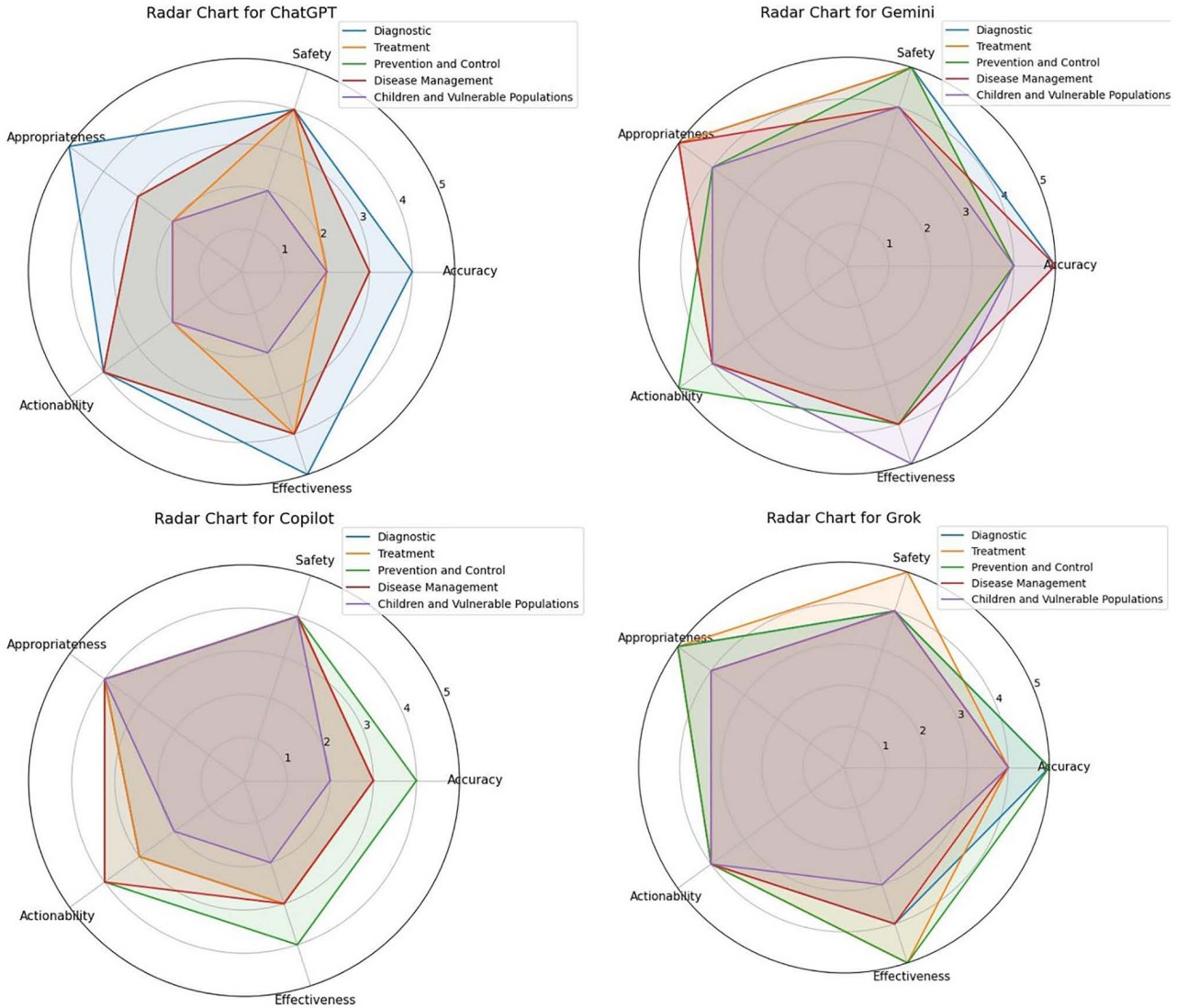

**Fig 1. The radar charts for the evaluation of the performance of four AI chatbots (ChatGPT, Gemini, Copilot, and Grok) based on the NLAT criteria (including Diagnostic, Treatment, Prevention and Control, Disease Management, and Children and Vulnerable Populations) and indicators such as Accuracy, Safety, Appropriateness, Actionability, and Effectiveness in the context of MS.**

The Grok model has a balanced and relatively high performance. Its chart is also large, but its scores may be slightly lower than Gemini's in some areas. This model performed particularly well in the Treatment and Prevention and Control domains, where it achieved maximum scores.

ChatGPT shows outstanding performance in some domains but weakness in others. For instance, in the Diagnostic and Disease Management domains, it performed very strongly on the Appropriateness and Effectiveness indicators. However, in the "Children and Vulnerable Populations" domain, its scores were consistently lower than the maximum across all indicators. This points to a more specialized rather than uniform performance.

In contrast, the Copilot model has the weakest performance among the four. Its radar chart covers the smallest area and did not reach the maximum score on any of the indicators. This shows that, compared to the other models, its performance was lower across all domains and criteria.

Regarding DISCERN, question level scores differed significantly across the four large language models (Kruskal–Wallis H (3) = 16.2, p = 0.001) (Tables 3). Dunn post hoc tests with Bonferroni correction showed that Gemini achieved significantly higher scores than ChatGPT (p_adj = 0.009) and Grok (p_adj = 0.002), whereas no other pairwise differences reached statistical significance (all p_adj > 0.05) (Tables 4).

Regarding NLAT, question level scores differed significantly across the four large language models (Kruskal–Wallis H(3) = 31.83, p < 0.00001) (Tables 3). Dunn post hoc tests with Bonferroni correction showed that Gemini achieved significantly higher scores than ChatGPT and Copilot, and Grok also outperformed ChatGPT and Copilot (Tables 4).

The data from the heatmap in Fig 2 indicate that Gemini and Grok achieved the highest scores in most domains. Gemini had the highest performance in the Diagnostic domain with an average score of 4.6, while Grok had the highest performance in the Children and Vulnerable Populations domain with an average score of 4.6. However, in the latter domain, Gemini had a lower performance with an average of 4.2, and Grok with an average of 3.8.

In contrast, Copilot and ChatGPT had lower scores than the other models in most domains. Specifically, in the "Children and Vulnerable Populations" domain, ChatGPT and Copilot had the lowest performance, with average scores of 2 and 2.8, respectively.

Overall, the color distribution in the heatmap of Fig 2 suggests that Gemini ranks first with a consistently high performance across all domains. Grok and ChatGPT follow closely with a small difference, while Copilot received the lowest overall score among the four models.

## 4. Discussion

This study aimed to evaluate and compare the performance of four major LLMs including ChatGPT, Gemini, Grok, and Microsoft Copilot, in responding to a curated set of clinically relevant questions about MS. Our findings shed light on the current capabilities, strengths, and limitations of LLMs in the context of medical information delivery, particularly for complex, chronic, and heterogeneous conditions such as MS. The analysis was conducted using two robust

**Table 3. Kruskal–Wallis tests for NLAT and DISCERN.**

| Outcome | H(3) | p-value |
|---|---|---|
| NLAT | 31.831 | < 0.00001 |
| DISCERN | 16.146 | 0.00106 |

**Table 4. Dunn's post hoc comparisons (Bonferroni-adjusted p-values).**

| Pair | NLAT p_ adjusted | DISCERN p_ adjusted |
|---|---|---|
| ChatGPT vs Copilot | 1.000000 | 0.588089 |
| ChatGPT vs Gemini | 0.000034 | 0.008620 |
| ChatGPT vs Grok | 0.001217 | 1.000000 |
| Copilot vs Gemini | 0.000191 | 0.752187 |
| Copilot vs Grok | 0.005106 | 0.249471 |
| Gemini vs Grok | 1.000000 | 0.002139 |

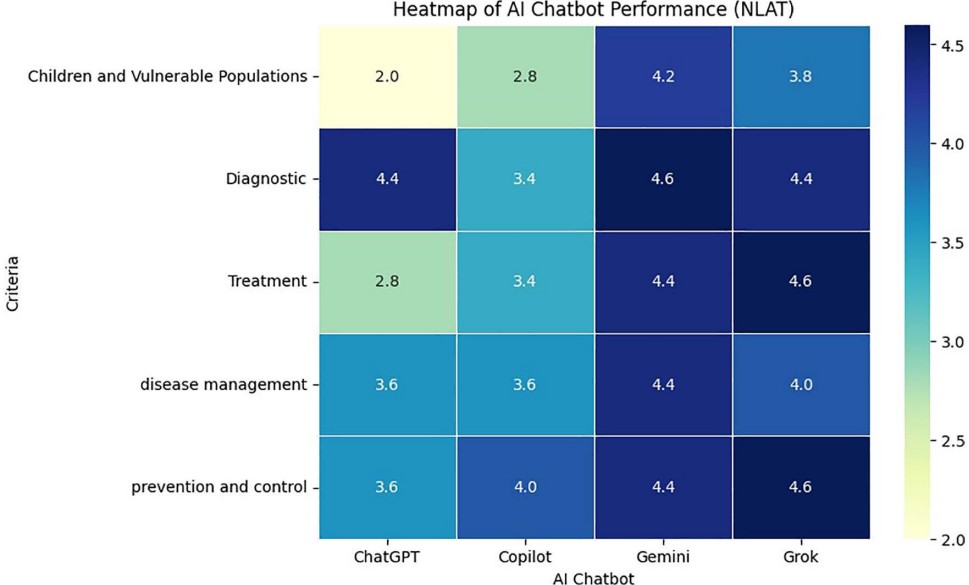

**Fig 2. A heatmap comparing the performance of four chatbots—ChatGPT, Copilot, Gemini, and Grok—in answering questions related to MS.**
The comparison is based on the average scores of the NLAT criteria across five main domains: Diagnostic, Treatment, Prevention and Control, Disease Management, and Children and Vulnerable Populations.

frameworks: DISCERN-AI criteria and NLAT (National Library of AI Tools) metrics, covering various performance indicators and clinical domains.

By combining a general information quality framework (DISCERN-AI) with a domain-specific clinical utility assessment (NLAT-AI), this study provides a layered evaluation strategy that more accurately reflects real-world use of LLMs in medical information seeking.

The findings of this study showed that among the evaluated models, Gemini demonstrated the most consistent overall performance in answering medical questions related to multiple sclerosis. It was the only model to receive an overall "high" quality rating according to the DISCERN-AI criteria and outperformed the others in areas such as relevance of content to the topic, transparency of sources, up-to-dateness of information, balance of content, and provision of supplementary resources. Moreover, the NLAT-AI results indicated that Gemini delivered uniformly strong performance—close to the maximum score—across all assessed domains and the indices of accuracy, safety, appropriateness, actionability, and effectiveness, reflecting a high level of stability and reliability for this model. These findings align with results of study by Agarwal et al. [25], which showed that Gemini and similar models (e.g., Claude 3.7) deliver high accuracy in medical MCQ evaluations. Notably, Gemini's superior performance across NLAT-AI's five core indicators (accuracy, safety, appropriateness, actionability, and effectiveness) across all evaluated domains reinforces its strength in translating medical knowledge into actionable, guideline-aligned content.

While ChatGPT did not lead in every category, it provided reliable, nuanced responses in complex domains reflecting moderate-to-high alignment with clinical guidelines. This echoes findings from Sahin Özdemir et al. [26], which observed ChatGPT's strength in structured, guideline-informed queries. This performance contrast is noteworthy given that Dabbas et al. [27] reported ChatGPT-3.5 achieving ~85% accuracy in neurolocalization tasks, indicating higher reliability in structured, well-defined neurological domains than in more heterogeneous conditions like MS. However, both ChatGPT and Gemini showed performance gaps in more complex or sensitive domains, consistent with results from study by Al-Thani et al [28], where final-year medical students outperformed LLMs in image-based and high-complexity cases, especially

in emergency medicine. Similarly, Chen et al. [29] found ChatGPT's performance to be domain-dependent, with strong results in general neurology questions but weaker accuracy in imaging, diagnostics, and critical care.

In contrast, Copilot emerged as the weakest performer, with substantial deficiencies in completeness, citation, and coverage of specialized topics like pediatric MS and maternal health. These limitations are consistent with previous reports suggesting that Copilot lags behind in clinical reasoning tasks when compared to more domain-trained models [30]. A critical concern in AI-generated medical content is the presence of hallucinations, fabricated or misleading statements, also raised by other recent study which noted hallucination and overconfidence issues in Gemini and Grok in complex diagnostic tasks [31]. These risks reinforce the need for human-in-the-loop supervision and caution against unfiltered use of LLM outputs in patient-facing applications.

Moreover, handling of uncertainty was inconsistent. While ChatGPT and Gemini showed better transparency by acknowledging evidence gaps or limitations in recommendations, Copilot frequently omitted qualifiers, which can mislead users about the certainty of a recommendation. Addressing this issue is essential to enhance user trust and maintain clinical safety.

This evaluation underscores the growing potential of LLMs as adjunct tools for patient education, pre-consultation support, and information dissemination in neurological care. As highlighted in recent work [32], their applications extend across MS care, research, and education, with particular promise in decision-support, monitoring, and integration with real-world data. Their greatest value lies in augmenting clinical practice—especially in settings with limited access to specialists or where patient self-education can be empowering—yet they cannot replace expert clinical judgment.

From an implementation perspective, Gemini and ChatGPT appear best positioned for integration into digital health platforms, provided they are regularly updated and governed by ethical AI frameworks. Prioritizing transparency (such as clear referencing and dating of information) is essential to foster safe and trustworthy use in healthcare settings.

Taken together, our findings, alongside prior studies, suggest that while LLMs are rapidly improving in medical dialogue generation, they still face challenges in handling specialized, sensitive, or evolving domains. Models like Gemini show particular promise for real-world applications in clinical decision support and patient education, but rigorous oversight, clinical validation, and careful prompt engineering remain critical for ensuring safe and effective use.

## 5. Strengths and limitations

Several limitations of the present study warrant consideration. First, the evaluation was conducted using static prompts, which do not capture the complexity of dynamic, multi-turn conversations that more accurately reflect real-world clinical interactions. Second, although the assessment was performed by an expert neurologist, reliance on a single reviewer limits the generalizability of the findings, as multiple evaluators would have enhanced inter-rater reliability. Third, the study was restricted to English-language prompts, raising questions about the applicability of these models to non-English-speaking populations or contexts requiring cultural and linguistic adaptation. Finally, it must be acknowledged that LLMs are continuously updated, and their performance may vary over time and across different versions.

## 6. Future directions

Looking ahead, several avenues for future research are evident. Longitudinal studies are needed to track model performance across updates and to evaluate outputs in multiple languages. Greater attention should also be directed toward domain-specific fine-tuning, including integration of structured clinical guidelines and patient-reported outcomes, to enhance medical relevance. In addition, the development of hybrid systems that combine human expertise with AI capabilities may provide a more reliable and ethically grounded framework for clinical use, particularly if they incorporate mechanisms for medical oversight and adaptive learning. Moreover, future studies should incorporate standardized hallucination detection frameworks to systematically evaluate factual inconsistencies in LLM responses. Another promising direction lies in exploring the role of LLMs in telehealth, where they may support real-time patient queries and follow-up triage.

Finally, dedicated ethical investigations remain critical to address issues of privacy, bias, misinformation, and liability, ensuring that the integration of LLMs into clinical encounters occurs in a safe, equitable, and trustworthy manner.

## 7. Conclusion

Based on the evaluation, Gemini demonstrated the strongest and most consistent overall performance across all assessed domains and criteria. Grok and ChatGPT showed moderate performance with domain-specific strengths, while Copilot performed the weakest across nearly all criteria. These findings highlight notable differences among large language models in generating accurate, reliable, and clinically actionable responses related to multiple sclerosis.

## Acknowledgments

We also appreciate the valuable assistance of colleagues and experts who provided feedback on the study design and interpretation of results. We also gratefully acknowledge the Social Development and Health Promotion Research Center of Gonabad University of Medical Sciences for their support.

## Author contributions

**Conceptualization:** Meisam Dastani, Mohammad Shayan Sajjadi, Bassem Yamout, Melika Arab Bafrani, Amirreza Nasirzadeh.

**Data curation:** Meisam Dastani, Mohammad Shayan Sajjadi, Bassem Yamout, Melika Arab Bafrani, Amirreza Nasirzadeh.

**Formal analysis:** Meisam Dastani, Bassem Yamout, Amirreza Nasirzadeh.

**Investigation:** Melika Arab Bafrani, Amirreza Nasirzadeh.

**Methodology:** Meisam Dastani, Bassem Yamout, Melika Arab Bafrani, Amirreza Nasirzadeh.

**Project administration:** Mohammad Shayan Sajjadi.

**Resources:** Mohammad Shayan Sajjadi, Bassem Yamout.

**Software:** Mohammad Shayan Sajjadi, Amirreza Nasirzadeh.

**Supervision:** Meisam Dastani, Melika Arab Bafrani, Amirreza Nasirzadeh.

**Validation:** Meisam Dastani, Bassem Yamout, Melika Arab Bafrani.

**Writing – original draft:** Mohammad Shayan Sajjadi, Bassem Yamout, Melika Arab Bafrani, Amirreza Nasirzadeh.

**Writing – review & editing:** Meisam Dastani, Mohammad Shayan Sajjadi, Bassem Yamout, Melika Arab Bafrani, Amirreza Nasirzadeh.

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
