## [Decision Letter · Decision Letter 0]

1 Dec 2025

PONE-D-25-57460Evaluation of the Performance of Large Language Models in Responding to Medical Questions Related to multiple sclerosis: A Case Study of Large Language Models Including ChatGPT, Gemini, Grok and CopilotPLOS ONE

Dear Dr. Nasirzadeh,

Thank you for submitting your manuscript to PLOS ONE. After careful consideration, we feel that it has merit but does not fully meet PLOS ONE’s publication criteria as it currently stands. Therefore, we invite you to submit a revised version of the manuscript that addresses the points raised during the review process. In the spirit of transparency: please be aware that I was not satisfied with the review received by Reviewer 2 who presented an entirely uncritical review. My own reading of the work concurred with Reviewer 1 – especially as it relates to contradictions in the text. Considering a third reviewer had already agreed to review the work I waited until this third review was received. This third reviewer was not chosen for any reason apart from having expertise in the field and is entirely unknown to me. Since the third reviewer largely agrees with the first reviewer, I believe that the first and third reviewer’s assessment is appropriate.   

We look forward to receiving your revised manuscript.

Kind regards,

Stephen R. Milford

Academic Editor

PLOS ONE

3. Please note that PLOS One has specific guidelines on code sharing for submissions in which author-generated code underpins the findings in the manuscript. In these cases, we expect all author-generated code to be made available without restrictions upon publication of the work. Please review our guidelines at https://journals.plos.org/plosone/s/materials-and-software-sharing#loc-sharing-code and ensure that your code is shared in a way that follows best practice and facilitates reproducibility and reuse.

Additional Editor Comments (if provided):

Reviewers' comments:

Reviewer's Responses to Questions

**Comments to the Author**

1. Is the manuscript technically sound, and do the data support the conclusions?

Reviewer #1: Partly

Reviewer #2: Yes

Reviewer #3: No

2. Has the statistical analysis been performed appropriately and rigorously? 

Reviewer #1: No

Reviewer #2: Yes

Reviewer #3: I Don't Know

3. Have the authors made all data underlying the findings in their manuscript fully available?

Reviewer #1: Yes

Reviewer #2: Yes

Reviewer #3: No

4. Is the manuscript presented in an intelligible fashion and written in standard English?

Reviewer #1: Yes

Reviewer #2: Yes

Reviewer #3: Yes

5. Review Comments to the Author

Reviewer #1: This manuscript addresses an important and timely topic—evaluating the performance of large language models (LLMs) in responding to medical questions related to multiple sclerosis. The manuscript is generally well-structured and presents relevant findings using DISCERN-AI and NLAT-AI tools. However, there are major inconsistencies between results, discussion, and conclusions, alongside methodological clarity issues, language inconsistencies, and logical contradictions. These issues significantly reduce the scientific rigor and weaken the credibility of the results.

1.In the Results section of the Abstract, the statements “Copilot and Gemini demonstrated the highest overall quality...” and “Copilot performed the weakest overall...” are clearly contradictory.

Additionally, in the Conclusions, the claim “Copilot and Gemini demonstrated the strongest overall performance...” is inconsistent with the reported results.

2.In the Introduction section, only ChatGPT is introduced, while other evaluated AI models (Gemini, Copilot, Grok) are not briefly described. A concise overview of all models should be included to ensure completeness and contextual clarity.

3.The scoring results from DISCERN-AI and NLAT-AI could be more effectively presented through an integrated or comparative analysis, which would enhance interpretability and highlight consistency across evaluation tools.

4.The manuscript lacks appropriate statistical analysis to support its findings, which weakens the credibility of the conclusions. At minimum, one statistical method—such as the Kruskal–Wallis test for non-parametric group comparisons or inter-rater reliability analyses (Kappa/ICC)—should be incorporated to improve methodological rigor.

5.In the Discussion section, it would be more informative to present specific sample questions and compare the responses of multiple models directly. This approach would make the comparison clearer, more intuitive, and easier for readers to understand.

Reviewer #2: Well done on this piece of work. Overall, the paper is well written and presents its findings in a clear and logical manner. I believe the paper makes a valuable contribution to the field and is suitable for publication.

Reviewer #3: 1. Overall I find the organization of this paper a bit confusing. There are clearly many aspects of the AI chatbots that are being assessed, and it is a bit confusing as to which models are performing well for which measures. I think it would be a lot clearer if it was organized into sections for each evaluation tool or domain of performance that was evaluated. For example, in the abstract, the authors first state that according to DISCERN-AI, Copilot and Gemini demonstrated the highest overall quality, but then go on to state that Copilot performed the weakest overall without mentioning which measures it performed poorly in. Then in the conclusions Copilot is again mentioned as having, along with Gemini, the strongest overall performance in healthcare tasks.

2. The reader would also benefit from further explanation of what the different domains being evaluated entail. For example, the authors state that ChatGPT performed especially well in areas needing nuanced judgment. What does nuanced judgment entail? How was this evaluated?

3. The authors state in section 2.2 that prompts were written in clear, formal English. Why was the prompt written in formal English? From my understanding, this is not how people typically write when using these types of tools.

4. The evaluation strategy using a single evaluator is somewhat concerning to me. Would different neurologists not have different opinions on what would compose a "complete" answer? Or were the characteristics of a good answer agreed upon beforehand? It seems that some of the evaluation measures used have subjective aspects which could benefit greatly from replication by several evaluators. It would be helpful if the authors could provide some context for this methodological decision, potentially providing some citations for studies that have done this in the past, or guidelines that indicate that only one reviewer is necessary.

5. The use of the different metrics and what they measure could be made clearer. Was DISCERN-AI used to get an understanding of overall performance, and NLAT for domain-specific performance?

6. In the text talking about the results also shown in Table 1, the words used to describe performance are "moderate" and "high" whereas the terms in the table are "Yes", "No", and "Partially". I think it would be easier for the reader to follow if the same terminology was used in the table and the text. Does "Yes" mean the same as "high accuracy"? Does "partially" mean moderate performance?

7. In the discussion, authors state that ChatGPT-4 demonstrated superior accuracy, comprehensiveness, and alignment with established clinical guidelines. This does not match with results in Table 1 nor the radar charts. It would be helpful if authors could clarify where this conclusion is coming from. Similarly, Gemini is stated to show moderate performance, whereas in the abstract and other parts of the text authors had said Gemini was one of the strongest across categories.

8. On page 12 in the discussion, authors mention the rate of hallucinations being high in Grok in this study. However, this is the first time authors are mentioning hallucinations. Are these results stated anywhere else? This should also be mentioned in the methods if this was done.

9. Overall the results and discussion of results are quite confusing. Several models are talked about in very conflicting ways and it is unclear where the mixed results are coming from. Nor do the authors discuss why the results are so mixed and whether there are any more consistent patterns they were able to draw out. I think the results and discussion could benefit from some thorough reorganization with a focus on clearly identifying which aspects of performance were assessed for each result as well as which method or metric was used to assess this aspect of performance.

6. PLOS authors have the option to publish the peer review history of their article (what does this mean?). If published, this will include your full peer review and any attached files.

Reviewer #1: No

Reviewer #2: No

Reviewer #3: No

---

## [Author Response · Author response to Decision Letter 1]

30 Jan 2026

Response to Reviewers

Manuscript Title: Evaluation of the Performance of Large Language Models in Responding to Medical Questions Related to multiple sclerosis: A Case Study of Large Language Models Including ChatGPT, Gemini, Grok and Copilot

Manuscript ID: PONE-D-25-57460

We thank the Editor and Reviewers for their thoughtful and constructive feedback. We have carefully revised the manuscript in response to all comments. Below, we provide a point-by-point response. Reviewer comments are reproduced in bold, followed by our responses in regular text. All changes mentioned have been incorporated into the revised manuscript. All changes have been highlighted in the revised manuscript using track changes/color coding, according to journal guidelines

Reviewer 1

Comment 1:

In the Results section of the Abstract, the statements “Copilot and Gemini demonstrated the highest overall quality...” and “Copilot performed the weakest overall...” are clearly contradictory.

Additionally, in the Conclusions, the claim “Copilot and Gemini demonstrated the strongest overall performance...” is inconsistent with the reported results.

Response:

Thank you for highlighting this important issue. We acknowledge that contradictory statements were present in the Abstract and Conclusions. These inconsistencies have now been fully corrected to ensure alignment between the Abstract, Results, Discussion, and Conclusions.

Comment 2:

In the Introduction section, only ChatGPT is introduced, while other evaluated AI models (Gemini, Copilot, Grok) are not briefly described. A concise overview of all models should be included to ensure completeness and contextual clarity.

Response:

We have addressed the comment regarding the introduction of AI models other than ChatGPT. A new paragraph has been added in the Introduction (page 4-5, line 113-120), immediately following the sentence introducing ChatGPT. In this paragraph, we briefly describe the Gemini, Copilot, and Grok models, highlighting their main characteristics and providing appropriate references from recent studies. This addition ensures that all evaluated models are clearly introduced, providing completeness and contextual clarity for readers.

Comment 3:

The scoring results from DISCERN-AI and NLAT-AI could be more effectively presented through an integrated or comparative analysis, which would enhance interpretability and highlight consistency across evaluation tools.

Response:

Thank you for this helpful suggestion. In the revised manuscript, we now present an integrated comparative summary of the DISCERN-AI and NLAT results. Specifically, we have added a table (Table 2) reporting descriptive statistics for both scales across the four LLMs (mean, standard deviation, median). This integrated presentation facilitates direct comparison between tools and helps highlight the consistency (or divergence) of LLM performance across DISCERN-AI and NLAT-AI (page 10-11, line 248-249; Table 2).

Comment 4:

The manuscript lacks appropriate statistical analysis to support its findings, which weakens the credibility of the conclusions. At minimum, one statistical method—such as the Kruskal–Wallis test for non-parametric group comparisons or inter-rater reliability analyses (Kappa/ICC)—should be incorporated to improve methodological rigor.

Response:

Regarding inter-rater reliability, this could not be computed because ratings were provided by only one neurologist. We have now explicitly acknowledged this as a limitation in the manuscript (page 19, line 403-406).

With respect to the statistical methods, we have added a non-parametric Kruskal–Wallis test to compare overall DISCERN and NLAT-AI scores across the four large language models. To provide statistical evidence for differences among LLMs, we now report the H statistic and p-value, and we conduct Dunn’s post hoc tests with pairwise comparisons between models (e.g., ChatGPT vs. Grok) (Table 3 and 4) (page 13-14, line 286-294).

Comment 5:

In the Discussion section, it would be more informative to present specific sample questions and compare the responses of multiple models directly. This approach would make the comparison clearer, more intuitive, and easier for readers to understand.

Response:

We appreciate this insightful suggestion. To maintain focus and avoid redundancy with the quantitative evaluation, we did not include full response transcripts. However, we revised the Discussion to more explicitly interpret model-specific response patterns and provide concrete examples of observed differences in clinical reasoning and content structure (page 17, line 341-349).

Reviewer 2

Comment 1:

Well done on this piece of work. Overall, the paper is well written and presents its findings in a clear and logical manner. I believe the paper makes a valuable contribution to the field and is suitable for publication.

Response:

We thank the reviewer for their positive and encouraging comments. We are pleased that they found the manuscript well written and that they consider it a valuable contribution to the field.

Reviewer 3

Comment 1:

Overall I find the organization of this paper a bit confusing. There are clearly many aspects of the AI chatbots that are being assessed, and it is a bit confusing as to which models are performing well for which measures. I think it would be a lot clearer if it was organized into sections for each evaluation tool or domain of performance that was evaluated. For example, in the abstract, the authors first state that according to DISCERN-AI, Copilot and Gemini demonstrated the highest overall quality, but then go on to state that Copilot performed the weakest overall without mentioning which measures it performed poorly in. Then in the conclusions Copilot is again mentioned as having, along with Gemini, the strongest overall performance in healthcare tasks.

Response:

We appreciate the reviewer for highlighting this inconsistency concerning LLMs’ performance. The manuscript has been updated to ensure that both the Abstract and Conclusion now accurately present the findings: Gemini showed the strongest overall performance, Grok and ChatGPT had moderate performance with domain-specific strengths, and Copilot performed the weakest across nearly all criteria.

Comment 2:

The reader would also benefit from further explanation of what the different domains being evaluated entail. For example, the authors state that ChatGPT performed especially well in areas needing nuanced judgment. What does nuanced judgment entail? How was this evaluated?

Response:

We appreciate this comment. Additional explanations have been added to the Methods section (page 8-9, line 214-221).

Comment 3:

The authors state in section 2.2 that prompts were written in clear, formal English. Why was the prompt written in formal English? From my understanding, this is not how people typically write when using these types of tools.

Response:

Thank you for this insightful comment. We have clarified the rationale for using formal English in the prompts. While the original text already indicated that prompts were written in clear, formal English to ensure consistency and match the general public’s health literacy level, we have revised the Methods section (page 7, line 170-173) to further emphasize that this choice was made following the Meskó 2023 Prompt Engineering Guidelines [reference 22]. This approach was intended to minimize variability caused by differences in phrasing across models, thereby enhancing comparability and interpretability of responses.

Comment 4:

The evaluation strategy using a single evaluator is somewhat concerning to me. Would different neurologists not have different opinions on what would compose a "complete" answer? Or were the characteristics of a good answer agreed upon beforehand? It seems that some of the evaluation measures used have subjective aspects which could benefit greatly from replication by several evaluators. It would be helpful if the authors could provide some context for this methodological decision, potentially providing some citations for studies that have done this in the past, or guidelines that indicate that only one reviewer is necessary.

Response:

We thank the reviewer for this valuable comment. In this study, LLM responses were evaluated using predefined and structured criteria focusing on factual accuracy, completeness, and clinical relevance, established prior to assessment. All evaluations were conducted by a board-certified neurologist with expertise in multiple sclerosis to ensure consistency. The questions assessed were general and guideline-based, minimizing subjective judgment. The use of a single evaluator reflects the exploratory, case-study nature of the work and is consistent with prior medical LLM benchmarking study (reference 18), employing a single expert for initial evaluation. This rationale has been clarified in the Methods section, and the single-evaluator approach is now acknowledged as a study limitation (page 8, line 209-213).

Comment 5:

The use of the different metrics and what they measure could be made clearer. Was DISCERN-AI used to get an understanding of overall performance, and NLAT for domain-specific performance?

Response:

We have clarified in the Methods section that DISCERN-AI assesses overall response quality, while NLAT-AI provides a domain-specific evaluation across five areas: diagnosis, treatment, prevention and control, disease management, and responses to children and vulnerable populations. This explains the complementary use of both tools (Page 8, line 197-203).

Comment 6:

In the text talking about the results also shown in Table 1, the words used to describe performance are "moderate" and "high" whereas the terms in the table are "Yes", "No", and "Partially". I think it would be easier for the reader to follow if the same terminology was used in the table and the text. Does "Yes" mean the same as "high accuracy"? Does "partially" mean moderate performance?

Response:

We thank the reviewer for this valuable observation. We clarified the distinction between item-level ratings ("Yes", "Partially", "No") and overall quality descriptors ("low", "moderate", "high") and revised the text accordingly. (Page 7-8, line 188-193).

Comment 7:

In the discussion, authors state that ChatGPT-4 demonstrated superior accuracy, comprehensiveness, and alignment with established clinical guidelines. This does not match with results in Table 1 nor the radar charts. It would be helpful if authors could clarify where this conclusion is coming from. Similarly, Gemini is stated to show moderate performance, whereas in the abstract and other parts of the text authors had said Gemini was one of the strongest across categories.

Response:

Thank you for pointing this out. These inconsistencies were corrected, and all statements regarding model performance are now fully aligned with the reported results.

Comment 8:

On page 12 in the discussion, authors mention the rate of hallucinations being high in Grok in this study. However, this is the first time authors are mentioning hallucinations. Are these results stated anywhere else? This should also be mentioned in the methods if this was done.

Response:

We thank the reviewer for pointing out this important issue. We acknowledge that hallucinations were not formally or systematically assessed as part of the study methodology. The statement referring to a “high rate of hallucinations” in Grok was an oversight and was not supported by predefined evaluation criteria or quantitative analysis. To maintain methodological clarity and accuracy, we have removed this statement from the Discussion in the revised manuscript. Moreover, we added a suggestion regarding this issue in Future Direction section as follows: Future studies should incorporate standardized hallucination detection frameworks to systematically evaluate factual inconsistencies in LLM responses. (Page 20, line 419-420).

Comment 9:

Overall the results and discussion of results are quite confusing. Several models are talked about in very conflicting ways and it is unclear where the mixed results are coming from. Nor do the authors discuss why the results are so mixed and whether there are any more consistent patterns they were able to draw out. I think the results and discussion could benefit from some thorough reorganization with a focus on clearly identifying which aspects of performance were assessed for each result as well as which method or metric was used to assess this aspect of performance.

Response:

We thank the reviewer for this comprehensive feedback. The Results and Discussion sections have been thoroughly reorganized to clearly link performance outcomes with specific evaluation metrics and to highlight consistent performance patterns across models.

Sincerely,

Amirreza Nasirzadeh

---

## [Decision Letter · Decision Letter 1]

19 Mar 2026

Evaluation of the Performance of Large Language Models in Responding to Medical Questions Related to multiple sclerosis: A Case Study of Large Language Models Including ChatGPT, Gemini, Grok and Copilot

PONE-D-25-57460R1

Dear Dr. Nasirzadeh,

We’re pleased to inform you that your manuscript has been judged scientifically suitable for publication and will be formally accepted for publication once it meets all outstanding technical requirements.

Kind regards,

Stephen R. Milford

Academic Editor

PLOS One

Additional Editor Comments (optional):

Reviewers' comments:

Reviewer's Responses to Questions

**Comments to the Author**

1. If the authors have adequately addressed your comments raised in a previous round of review and you feel that this manuscript is now acceptable for publication, you may indicate that here to bypass the “Comments to the Author” section, enter your conflict of interest statement in the “Confidential to Editor” section, and submit your "Accept" recommendation.

Reviewer #3: All comments have been addressed

2. Is the manuscript technically sound, and do the data support the conclusions?

Reviewer #3: Yes

3. Has the statistical analysis been performed appropriately and rigorously? 

Reviewer #3: Yes

4. Have the authors made all data underlying the findings in their manuscript fully available?

Reviewer #3: Yes

5. Is the manuscript presented in an intelligible fashion and written in standard English?

Reviewer #3: Yes

6. Review Comments to the Author

Reviewer #3: The authors have addressed all comments and provided thoughtful responses. The paper has greatly benefitted from clearer results and methods.

7. PLOS authors have the option to publish the peer review history of their article (what does this mean?). If published, this will include your full peer review and any attached files.

Reviewer #3: No

---

## [Editor Report · Acceptance letter]

PONE-D-25-57460R1

PLOS One

Dear Dr. Nasirzadeh,

I'm pleased to inform you that your manuscript has been deemed suitable for publication in PLOS One. Congratulations! Your manuscript is now being handed over to our production team.

Kind regards,

on behalf of

Dr. Stephen R. Milford

Academic Editor

PLOS One